# Garlic Consumption and All-Cause Mortality among Chinese Oldest-Old Individuals: A Population-Based Cohort Study

**DOI:** 10.3390/nu11071504

**Published:** 2019-06-30

**Authors:** Xiaoming Shi, Yuebin Lv, Chen Mao, Jinqiu Yuan, Zhaoxue Yin, Xiang Gao, Zuofeng Zhang

**Affiliations:** 1National Institute of Environmental Health, Chinese Center for Disease Control and Prevention, Beijing 100021, China; 2Division of Epidemiology, School of Public Health, Southern Medical University, Guangzhou 510515, China; 3Division of Non-Communicable Disease Control and Community Health, Chinese Center for Disease Control and Prevention, Beijing 102206, China; 4Department of Nutritional Sciences, The Pennsylvania State University, University Park, PA 16802, USA; 5Department of Epidemiology, Fielding School of Public Health, University of California, Los Angeles (UCLA), Los Angeles, CA 90095, USA

**Keywords:** garlic, mortality, survival, oldest old, elderly

## Abstract

In vitro and in vivo experimental studies have shown garlic has protective effects on the aging process; however, there is no evidence that garlic consumption is associated with all-cause mortality among oldest-old individuals (≥80 years). From 1998 to 2011, 27,437 oldest-old participants (mean age: 92.9 years) were recruited from 23 provinces in China. The frequencies of garlic consumption at baseline and at age 60 were collected. Cox proportional hazards models adjusted for potential covariates were constructed to estimate hazard ratios (HRs) relating garlic consumption to all-cause mortality. Among 92,505 person-years of follow-up from baseline to September 1, 2014, 22,321 participants died. Participants who often (≥5 times/week) or occasionally (1–4 times/week) consumed garlic survived longer than those who rarely (less than once/week) consumed it (*p* < 0.001). Participants who consumed garlic occasionally or often had a lower risk for mortality than those who rarely consumed garlic at baseline; the adjusted HRs for mortality were 0.92(0.89–0.94) and 0.89(0.85–0.92), respectively. The inverse associations between garlic consumption and all-cause mortality were robust in sensitivity analyses and subgroup analyses. In this study, habitual consumption of garlic was associated with a lower all-cause mortality risk; this advocates further investigation into garlic consumption for promoting longevity.

## 1. Introduction

Used universally for flavoring and in traditional medicines and functional foods to enhance physical and mental health, garlic (*Allium sativum*) is among the most popular condiments worldwide [1]. Garlic contains organosulfur compounds (OSCs), which largely account for its protective effects [2]. It has been suggested that garlic can prevent cardiovascular disease, cancer, and diseases associated with aging by in vitro and in vivo studies [3]. In population-based studies, a number of beneficial effects have long been documented for garlic or its bioactive ingredients, especially the prevention of cancer and cardiovascular disease [4,5]. Case–control studies also documented the protective effects of garlic consumption in gastric, colorectal, prostate, head and neck, and lung cancers in different populations [4,6,7,8]. Evidence from randomized controlled trials indicated that garlic is beneficial in treating hypertension [9]. However, the association between garlic supplement intake with gastric, colorectal, breast, or lung cancers in prospective cohort studies has not been found [10]. Systematic reviews of randomized controlled trials have concluded that the inverse relationship between garlic supplementation and cardiovascular system disorders was only observed in special populations [11]. There is also insufficient evidence to determine the difference in the reduction of cardiovascular morbidity risk and mortality risk between patients treated with garlic supplements or placebo [12].

Reactive oxygen species (ROSs) accumulate in the aging process. The free radical theory has been proposed to explain the phenomenon of aging. Garlic (especially its OSCs) is known to exhibit antioxidant properties and is the most commonly used herbal supplement among community-dwelling elderly [13]. However, there is a lack of epidemiologic evidence on the protective effects of garlic against morbidity or mortality in the elderly. In China, garlic is consumed mainly in its raw form, and its consumption is much higher than elsewhere in the world [4,14]. Better biologic properties have been suggested for raw garlic than for cooked (processed or heated) garlic: heat seems to react with certain substances in garlic and change their chemical composition [15]. As the largest exporter and producer of garlic [16], the Chinese population would appear to be well suited for studying the association of garlic intake with mortality.

Uncertainties about the association between intake of either garlic or garlic supplements and health outcomes could be the result of confounding factors (e.g., tobacco smoking, alcohol drinking, tea drinking, or other dietary habits) or methodological issues (e.g., insufficient sample sizes). Therefore, to evaluate the association of dietary garlic consumption with all-cause mortality, we assessed information from 27,437 oldest-old individuals (≥80 years), adjusting for potential confounding factors, including demographic status, socioeconomic status, dietary habits, health practices, comorbidity, cognition status, and activities of daily living (ADL).

## 2. Materials and Methods

### 2.1. Study Design and Participants

The Chinese Longitudinal Healthy Longevity Study (CLHLS) was conducted in 23 Chinese provinces using multi-stage stratified sampling. The study has one of the world’s largest samples specifically focusing on oldest-old individuals; the population in the survey areas constitutes about 85% of the total population in China. Participants were enrolled in 1998, 2000, 2002, 2005, 2008–09 and 2011–12; they have been followed up ever since in this dynamic cohort. A detailed description of the CLHLS appears elsewhere [17,18,19]. All procedures performed in studies were approved by the biomedical ethics committee of Peking University (IRB00001052-13074). Written informed consents were obtained from all participants included in the study, either themselves or their proxy respondents.

Briefly, 43,487 participants had valid baseline data, i.e., they completed the questionnaire and physical examination measurements. To meet the inclusion criteria, we excluded 9131 participants aged 79 years or younger, 5019 participants lost to follow-up at the first follow-up survey, and 1900 participants with missing garlic consumption information. Thus, the final analyses covered 27,437 participants: 9496 octogenarians, 9475 nonagenarians, and 8466 centenarians (In Appendix A). Various characteristics, i.e., garlic consumption, age, sex, and diseases and functional status, were compared between participants lost-to-follow-up or not at the first follow-up survey. A significant but relatively petty difference was found for garlic consumption (17.5% vs. 15.0% often consumed garlic; 35.3% vs. 35.1% occasionally consumed garlic; 47.1% vs. 49.9% rarely consumed garlic; *p* < 0.001), age (94.1 vs. 92.7 years; *p* < 0.001), hypertension (54.8% vs. 52.9%; *p* = 0.013), and cognitive impairment (34.0% vs. 37.1%; *p* < 0.001) between the two groups, while no significant difference was found for sex (37.7% vs. 39.0% men; *p* = 0.080) and respiratory disease (11.2% vs. 11.9%; *p* = 0.154).

### 2.2. Assessment of Garlic Consumption

In the baseline survey, garlic consumption habits were determined using a brief food frequency questionnaire that has been shown to be effective in assessing the risk of functional performance and mortality [18,20,21]. Participants were asked two questions: “How often do you currently consume garlic?” and “How often did you consume garlic when aged 60 years?”. As all participants enrolled in this study were older than 60, the frequency of garlic consumption was collected through retrospective investigation. The questions were in the form of a qualitative food frequency questionnaire. Three response categories were chosen: often (daily or almost daily, ≥5 times/week), occasionally (1–4 times/week), and rarely (less than once/week) [18]. It should be noted that only the frequency of garlic consumption was determined, and no quantitative assessment was made. In the present study, we used baseline garlic consumption as primary exposure. We also explored the association between garlic consumption at age 60 years and mortality in a secondary analysis. To test the potential influence of recall bias, the correlation between garlic consumption at baseline and at age 60 years was analyzed and verified with good consistency (Pearson correlation coefficient = 0.79), which also indicates that the habit of garlic consumption was relative stable among Chinese elderly aged 60 and older.

### 2.3. Death Ascertainment

Information about death was obtained from participants’ close family members or local residential committees in the follow-up survey [18]. Person-years were calculated from baseline to the date of loss to follow-up, death, or September 1, 2014, whichever occurred first. We defined the survival time for participants as the period from recruitment to death. In this study, participants who could not be contacted in the further follow-up survey were defined as “lost to follow-up”.

### 2.4. Statistical Analysis

There were missing values in 11.6% of body mass index (BMI) data and less than 2% of other independent variables. A multiple-imputation approach was adopted to reduce the influence of missing values on the models [22]. Octogenarians, nonagenarians, and centenarians have different risks of morbidity and mortality; thus, we performed analyses among the three age groups separately. Covariates, including demographic variables, socioeconomic status, dietary behaviors, collected using the brief food frequency questionnaire [18,20,21], health practices, functional performance measured using the Katz scale and the Chinese version of the mini-mental state examination (MMSE) were considered for analysis [23,24,25]. On the basis of previous findings in epidemiologic or clinical studies, in our data analyses we adjusted for a number of confounding factors associated with mortality or garlic consumption. All the CLHLS data were collected by a well-trained research team member and a local county doctor or nurse using a structured questionnaire in a face-to-face interview. If a respondent was unable to answer questions owing to illness or cognitive limitations, the next of kin, a family member, or a very close friend or caregiver responded instead [18].

Demographic variables included age (as a continuous variable), sex (men or women), and residence (urban or rural). Socioeconomic status included educational background (literacy or illiteracy), occupation (farmer or other), marital status (in marriage or not in marriage), and living pattern (with family members, alone or at nursing home). Dietary behaviors, such as vegetable consumption (often, occasionally, or rarely), fruits consumption (often, occasionally, or rarely), meat consumption (often, occasionally, or rarely), fish consumption (often, occasionally, or rarely), and tea drinking (often, occasionally, or rarely) were collected using the brief food frequency questionnaire [18,20,21] and were included in the adjusted model. Health practices were assessed in terms of tobacco smoking status (current smoker, former smoker, or non-smoker), alcohol drinking status (current drinker, former drinker, or non-drinker), and currently undertaking regular exercise (yes or no).

Collected data on medical history included heart disease, cerebrovascular disease, and respiratory disease. Hypertension was defined as systolic blood pressure ≥140 mmHg or diastolic blood pressure ≥90 mmHg. Functional performance, including ADL and cognitive function, was also assessed. ADL disability was measured using the Katz scale [22]: having difficulty in performing one or more of six daily activities (bathing, dressing, eating, indoor transfers, toileting, and continence). Cognitive function was assessed using the Chinese version of MMSE, which tests the following functioning aspects: orientation, calculation, recall, and language [23]. Cognitive impairment was defined by an MMSE score of less than 18 out of 30 [24]. Among the three age groups, we determined the differences in covariates according to garlic consumption using the Cochran–Mantel–Haenszel test and one-way analysis of variance. BMI (as continuous variable) was computed as weight/squared body height in units of kg/m^2^. Height was measured to the nearest 1 centimeter (cm) for participants in the 2005, 2008, and 2011 surveys or was estimated according to knee height (in cm) with a validated equation for participants in the 1998, 2000, and 2002 surveys as follows: body height for men = 2.01 × knee height + 67.78, body height for women = 1.81 × knee height + 74.08 [26,27]. After adjusting for the confounding factors above, Cox proportional hazards regression model was constructed to calculate the hazard ratios (HRs) and 95% confidence intervals of mortality for garlic consumption.

To explore the robustness of the association, a number of sensitivity analyses were performed. To determine whether chronic disease (especially of the digestive system) might be associated with lack of garlic consumption, we adjusted for digestive system disease (such as gastrointestinal ulcers, chronic hepatitis or cirrhosis, and gallstones or cholecystitis), income (very rich, rich, so-so, poor, very poor), insurance (yes or no), and vitamins A, C, and E. We excluded participants who had heart disease, cerebrovascular disease, or respiratory disease. We also excluded participants who died during the first year of follow-up.

Several subgroup analyses were conducted to investigate the association between garlic consumption and all-cause mortality on the basis of the following parameters: sex (men and women); tobacco smoking status (current or former smoker, or non-smoker); alcohol drinking status (current or former alcohol drinker, or non-drinker); tea drinking status (drinker or non-drinker); ADL disability (yes or no), and cognitive function (impaired or normal) [23,24,25]. We performed multiplicative interaction treating garlic consumption as a continuous variable (garlic consumption frequency, valued as 1, 2, 3 for rarely, occasionally, and often, respectively) in the fully adjusted model. We performed the statistical analyses using SAS 9.4 (SAS Institute, Inc., Cary, NC, USA) and R software for Windows (version 3.5.0; R Foundation for Statistical Computing, Vienna, Austria).

## 3. Results

### 3.1. Garlic Consumption and Baseline Characteristics

Table 1 presents the baseline characteristics of the participants in terms of garlic consumption. Among the 27,437 participants in this study, the mean age was 92.9 years; 61.7% of the participants were women, and 75.0% lived in rural areas (Table 1). Overall, 15.9% of participants reported often consumed garlic. Garlic consumption decreased with age: it was reported by 17.6% of octogenarians and 15.0% of nonagenarians and centenarians. The participants who consumed garlic more frequently were more likely to be men, married, literate, living with family members, current smokers, current or former alcohol consumers, and more frequently consumed meat, vegetables, fruits, and tea. Notably, the participants who consumed garlic more frequently were less likely to be hypertensive and cognitively impaired (Table 1). 

### 3.2. Garlic Consumption at Baseline or at 60 Years of Age and All-Cause Mortality

Among 92,505 person-years of follow-up from baseline to 2014, we documented 22,321 deaths. The participants who often consumed garlic survived longer than those who occasionally or rarely consumed it: the median of survival time was 3.2, 3.0, and 2.7 years, respectively (*p* < 0.001). Greater garlic consumption was associated with lower all-cause mortality after further adjustment for potential confounders. For the whole cohort, compared with participants who rarely consumed garlic at baseline, the adjusted HRs for mortality were 0.92 (0.89–0.94) and 0.89 (0.85–0.92) for those who consumed garlic occasionally (1–4 times/week) or often (≥5 times/week), respectively (Figure 1a and Figure 2). The HRs were 0.96 (0.93–0.98) and 0.78 (0.75–0.82), respectively, for participants who occasionally or often consumed garlic compared with those who rarely consumed it at the age of 60 years (Figure 1b and Figure 3).

### 3.3. Sensitivity Analyses and Subgroup Analyses

In the sensitivity analyses, the association of garlic consumption at present or at age 60 with all-cause mortality remained robust when additionally adjusted for digestive system disease. Likewise, that association remained unchanged after further adjustments for insurance, income, and vitamins A, C, and E. The association did not change after excluding participants with heart disease, cerebrovascular disease, or respiratory disease at baseline. The association was similarly unchanged after excluding participants who died during the first year of follow-up. The subgroup analysis for the association between garlic consumption and all-cause mortality indicated that the protective effect of garlic consumption persisted among the following participants: octogenarians, nonagenarians, and centenarians; men and women; current or former smokers and non-smokers; current or former alcohol drinkers and non-drinkers; tea drinkers and non-drinkers; participants with cognitive impairment and normal cognition; and participants with disability and normal ADL. Notably, the protective effect of garlic consumption on survival was modified by age. The inverse association between mortality and garlic consumption at baseline or at 60 years was more prominent in octogenarians and nonagenarians (*p* for interactions were 0.018 and <0.001), in participants rarely drinking tea (*p* for interaction were <0.001), in participants with normal cognition (*p* for interaction were 0.027 and 0.022). The inverse association between mortality and garlic consumption at baseline was more prominent in men (*p* for interactions was 0.035).

## 4. Discussion

It was found that the habit of garlic consumption was inversely associated with mortality among 27,437 participants aged over 80 years in China. The oldest old who consumed garlic more than five times a week had a 11% decrease in the risk of mortality, compared with those who consumed it less than once a week. The association was consistent across different age groups (octogenarians, nonagenarians, centenarians), for different sex, and in different participant subgroups defined by various baseline characteristics.

This is the first study, to our knowledge, to examine the association between daily garlic consumption and mortality among the oldest old in a prospective cohort. A possible explanation for the results is reverse causality: participants with comorbidities (especially digestive system disease) might limit the consumption of garlic. However, the association remained after we excluded, in the sensitivity analyses, participants with comorbidities to eliminate a potential bias caused by a dietary change due to preexisting illness, affecting the exposure-outcome relation in this cohort study. Moreover, after excluding participants who died during the first year or further adjusting for the presence of digestive system diseases, we found that the association was robust.

The present result is in accordance with former studies revealing beneficial impacts of garlic consumption. Garlic contains a number of potentially active chemical constituents: OSCs, enzymes, amino acids and their glycosides, Se, and other trace minerals [28]. Almost all studies have focused on OSCs as the active principles of garlic [3], which in experimental or small-population studies have been demonstrated to be largely responsible for garlic therapeutic properties [29,30]. However, other components may contribute to its overall beneficial effects [31]. The biological activities of garlic (including anticarcinogenic, antioxidant, antibacterial, anti-inflammatory, antiprotozoal, antifungal, hypotensive, hypolipidemic, and antidiabetic actions) have been extensively investigated [3]. Epidemiologically, the consumption of garlic was linked to a lower risk of colon, stomach, and oral cancers [4,6,7,8]. Clinically, garlic supplements are used as hypotensives, cholesterol-lowering drugs, as well as glucose-lowering drugs [9,11,12]. Although the effects of garlic’s bioactive ingredients on human health and the aging process require further investigation, some possible explanations have been proposed to explain the benefits of garlic consumption in the oldest old.

To explain the phenomenon of aging, many theories have been advanced, among which the free radical theory is widely accepted. It is generally acknowledged that with age, ROSs are generated, and oxidative damage to human cells accumulates; the damage is likely to contribute to degenerative diseases in the aging process [32,33]. Almost all the above-mentioned chemical constituents of garlic, especially OSCs, have been shown to possess antioxidant properties both in vivo and in vitro [34]. Thereby, these constituents possibly mediate the anti-aging effects of garlic. Accordingly, evidence from a few interventional studies in humans has documented the antioxidant properties of garlic. Plasma oxidized low-density lipoprotein (OxLDL) levels were found to fall sharply in subjects who received garlic supplementation compared with controls [35]. Altogether, garlic and its OSCs have the potential to delay the onset and development of aging through suppressing the expression of NF-κB and lowering ROS concentration [36].

Some other potential explanations for the protective effects of garlic consumption on survival have been advanced, as described below. (1) Garlic and cancer: A study by the US Food and Drug Administration and a Chinese double-blind intervention study have revealed the anti-carcinogenic potential of garlic [37,38]. A garlic extract has been reported to inhibit the growth of several different cancer cells in vitro, as well as cancer growth in vivo, enhancing the activities of chemotherapeutics, and of MAPK and PI3K inhibitors [39]. Besides, similar to other substances like olive oil, garlic may also have a chemopreventive action due to its phenolic compounds [40]. (2) Garlic and cardiovascular disease (CVD): Garlic is able to enhance the production of hydrogen sulfide, reduce the bioavailability of nitric oxide, inhibit platelet aggregation, and lower hypercholesterolemia and hypertension, thereby potentially preventing CVD [41,42,43]. (3) Antidotal effect of garlic: Both sulfur atoms and allyl groups in OSCs are vital to phase I and II transcription enzymes, which detoxify various endogenous and exogenous chemicals and accelerate metabolite excretion rate [41]. (4) Anti-inflammatory function of garlic: Garlic has been used as an antibacterial and an immunity stimulant [44]; its use is well reported in treating ear infections, toothaches, and upper respiratory infections [45].

The present study found that garlic consumption, especially ≥5 times a week, was associated with lower mortality risk, possibly due to the bioaccessibility and bioavailability of garlic’s chemical components. This study found that the protective effect was modified by age. The physiological function of the digestive system declines with age; for centenarians, it is less easy to absorb the effective components (such as OSCs) of garlic, which leads to garlic’s weaker protective effect on survival in this age group. Dietary behavior and appetite are highly correlated with global cognitive function. Thus, it was unsurprising that garlic consumption was particularly associated with reduced mortality risk in people with normal cognition, who had a healthier dietary behavior and better health status [46,47]. Further targeted investigations are warranted to explore the influence of potential interactions of garlic with other factors on all-cause mortality, especially tea drinking and gender.

Regarded as a safe food, garlic has been consumed for a long time. However, several health risks are reportedly associated with excessive consumption of garlic. It has been proposed that garlic may interact with some drugs and reduce their efficacy [48]. Particularly, allergic reactions, gastrointestinal tract injury, and decrease in body weigh caused by garlic have attracted much attention [41,48]. Accordingly, it would seem prudent for individuals with digestive system disease under medication to avoid garlic.

This study has some limitations. First, the quantity of garlic consumption was not assessed, preventing us from making definite conclusions about a dose–response relationship between the quantity of garlic consumed and the mortality rate; therefore, it not possible to provide more detailed advice on garlic consumption. Second, exposure of garlic to high temperatures was found to destroy OSCs and inhibit garlic anticancer activity [49]. We did not obtain information about how garlic was consumed (raw, cooked, or processed); however, in China—unlike in other countries—garlic is mainly consumed raw [4,14]. Third, the total energy intake was not adjusted for in our study, even though we adjusted for many other variables closely related to it, including age, sex, regular exercise, and BMI. Fourth, the overall dietary patterns were not adjusted for; garlic consumption in our study was closely correlated with other dietary habits, such as consumption of meat, fish, vegetables, and tea; garlic consumption may be an indicator of good appetite, which in turn has been closely associated with health status and risk of all-cause mortality [47]. Although we carefully adjusted for some dietary habits and conducted a comprehensive sensitivity analysis obtaining robust findings, residual confounding factors, such as intake of minerals, hypercholesterolemia, medication use, can still be present. Fifth, cause-specific mortality was not documented for the participants, especially death of cancer or CVD. Further studies on the anti-cancer and CVD prevention effect of garlic consumption are recommended. Sixth, a recall bias in the collection of garlic consumption at the age of 60 may exist, even though the habit of garlic consumption was relative stable at baseline and at the age of 60 for this population. Seventh, due to the exclusion of 5019 participants lost-to-follow-up at the first follow-up survey, a selection bias may confound the findings, even though the differences between included and excluded subjects were very small.

## 5. Conclusions

In summary, garlic consumption was associated with a reduced risk of all-cause mortality. The present finding provides a limited evidence on the association of garlic with all-cause mortality among the oldest old; it advocates future research into the dose–response relationship between the quantity of garlic consumption and cause-specific mortality, especially for CVD and cancer mortality. It would seem beneficial to promote garlic intake as part of a healthy daily diet among octogenarians, nonagenarians, men, people with normal cognition, and people who rarely drink tea. If these associations are largely causal, the potential health gain from increased garlic consumption in China would be substantial on the basis of a nationally representative sample. The garlic constituents can be further studied to identify effective agents for inflammation and cancer treatment and CVD prevention. However, to verify the role of garlic or garlic supplements in longevity, our results need to be confirmed by future prospective human studies and community intervention trials.

## Figures and Tables

**Figure 1 nutrients-11-01504-f001:**
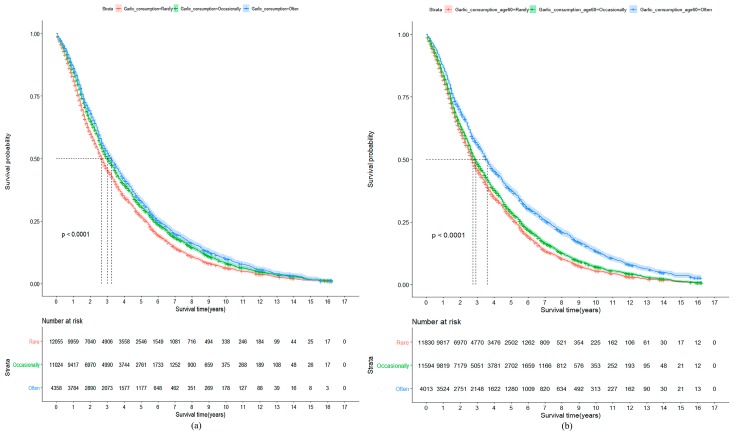
(**a**) Kaplan–Meier plot showing mortality according to garlic consumption at baseline. Participants who often or occasionally consumed garlic at baseline survived longer than those who rarely consumed it (median survival time was 3.2, 3.0, and 2.7 years, respectively; *p* < 0.001). (**b**) Kaplan–Meier plot showing mortality according to garlic consumption at age 60. Participants who often consumed garlic at age 60 survived approximately one year longer than those who occasionally or rarely consumed it (median survival time was 3.7, 2.9, and 2.8 years, respectively; *p* < 0.001).

**Figure 2 nutrients-11-01504-f002:**
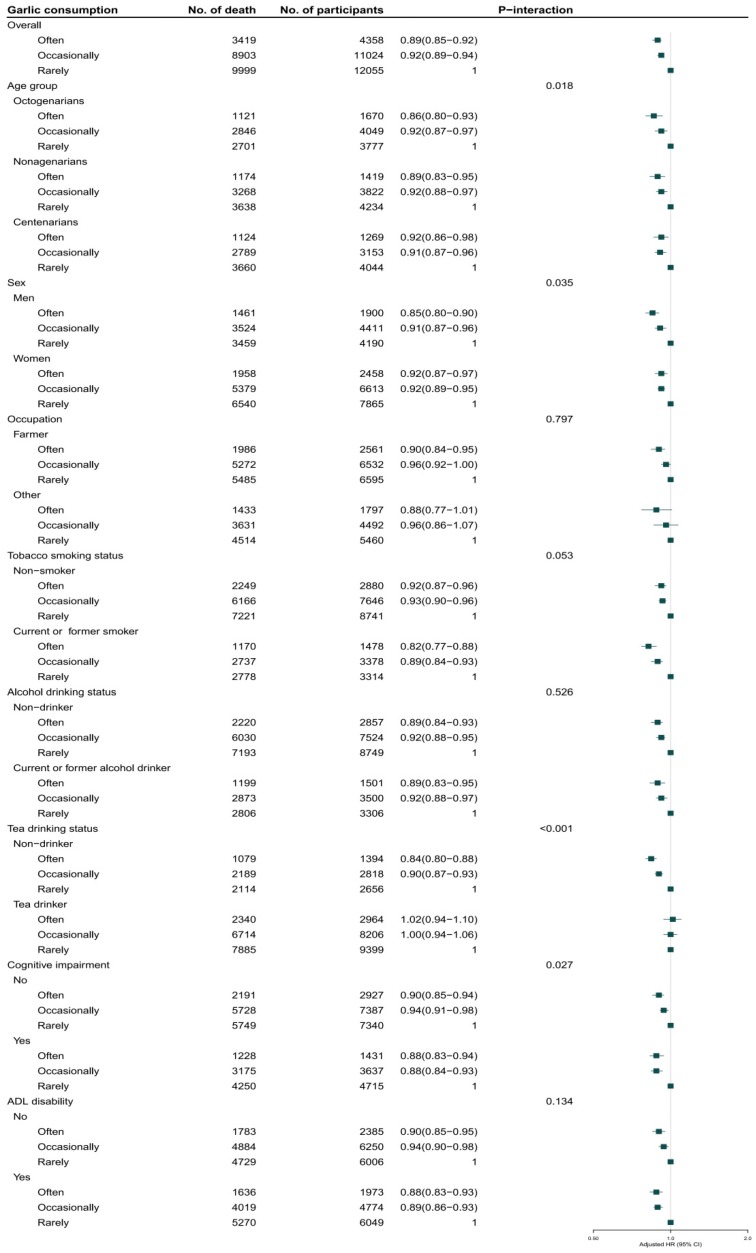
Association of garlic consumption at baseline with all-cause mortality in the oldest old. Adjusted for age, sex, BMI, residence, educational background, occupation, current marital status, living pattern, tobacco smoking, alcohol drinking, regular exercise, eating vegetables, eating fruits, meat, and fish, drinking tea, hypertension, heart disease, cerebrovascular disease, respiratory disease, cognitive impairment, and ADL disability.

**Figure 3 nutrients-11-01504-f003:**
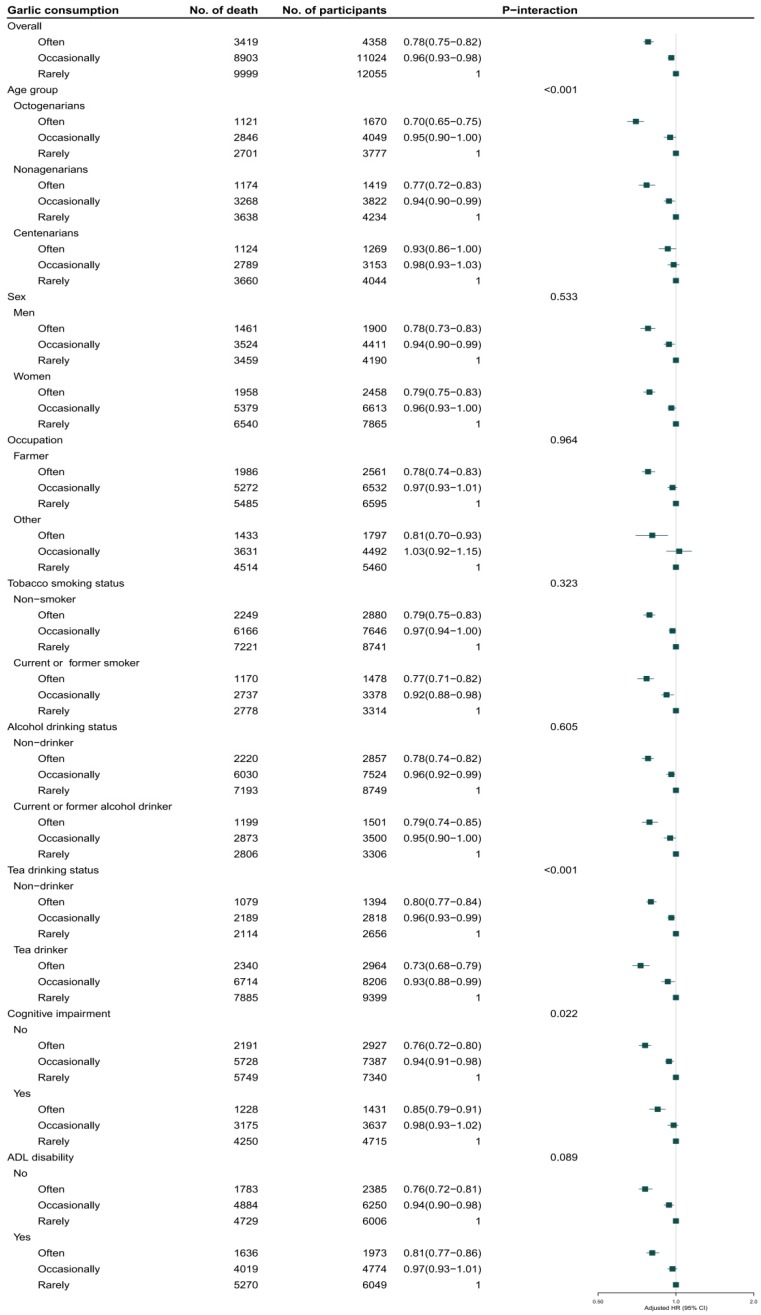
Association of garlic consumption at age 60 with all-cause mortality in the oldest old. Adjusted for age, sex, BMI, residence, educational background, occupation, current marital status, living pattern, tobacco smoking, alcohol drinking, regular exercise, eating vegetables, eating fruits, meat, and fish, drinking tea, hypertension, heart disease, cerebrovascular disease, respiratory disease, cognitive impairment, and ADL disability.

**Table 1 nutrients-11-01504-t001:** Baseline characteristics of the study participants according to garlic consumption by octogenarians, nonagenarians, and centenarians.

TbVariable	Octogenarians				Nonagenarians				Centenarians				All			
	Often	Occasionally	Rare	*p* Value	Often	Occasionally	Rare	*p* Value	Often	Occasionally	Rare	*p* Value	Often	Occasionally	Rare	*p* Value
**Number of participants**	1670 (17.6)	4049 (42.6)	3777 (39.8)		1419 (15.0)	3822 (40.3)	4234 (44.7)		1269 (15.0)	3153 (37.2)	4044 (47.8)		4358 (15.9)	11024 (40.2)	12055 (43.9)	
**Age (year)**	84.3 ± 3.0	84.2 ± 3.0	84.3 ± 3.0	0.746	93.5 ± 2.8	93.7 ± 2.9	93.8 ± 2.9	0.001	101.6 ± 2.0	101.6 ± 2.0	101.5 ± 1.9	0.003	92.3 ± 7.6	92.5 ± 7.5	93.4 ± 7.4	<0.001
**BMI (kg/m^2^)**	20.1 ± 3.8	19.8 ± 3.6	19.7 ± 3.9	0.001	19.4 ± 3.7	18.9 ± 3.3	18.9 ± 3.3	<0.001	18.7 ± 3.4	18.5 ± 3.3	18.4 ± 3.3	0.001	19.5 ± 3.7	19.1 ± 3.5	19.0 ± 3.5	<0.001
**Sex**				<0.001				<0.001				0.089				<0.001
**Men**	932 (55.8)	2127 (52.5)	1770 (46.9)		693 (48.8)	1664 (43.5)	1657 (39.1)		275 (21.7)	620 (19.7)	763 (18.9)		1900 (43.6)	4411 (40.0)	4190 (34.8)	
**Women**	738 (44.2)	1922 (47.5)	2007 (53.1)		726 (51.2)	2158 (56.5)	2577 (60.9)		994 (78.3)	2533 (80.3)	3281 (81.1)		2458 (56.4)	6613 (60.0)	7865 (65.2)	
**Occupation**				0.019				0.006				<0.001				<0.001
**Farmer**	923 (55.3)	2228 (55.0)	1970 (52.2)		811 (57.2)	2253 (58.9)	2347 (55.4)		827 (65.2)	2051 (65.0)	2278 (56.3)		2561 (58.8)	6532 (59.3)	6595 (54.7)	
**Other**	747 (44.7)	1821 (45.0)	1807 (47.8)		608 (42.8)	1569 (41.1)	1887 (44.6)		442 (34.8)	1102 (35.0)	1766 (43.7)		1797 (41.2)	4492 (40.7)	5460 (45.3)	
**Residence**				<0.001				<0.001				<0.001				<0.001
**Urban**	457 (27.4)	1007 (24.9)	1129 (29.9)		340 (24.0)	782 (20.5)	1146 (27.1)		270 (21.3)	628 (19.9)	1099 (27.2)		1067 (24.5)	2417 (21.9)	3374 (28.0)	
**Rural**	1213 (72.6)	3042 (75.1)	2648 (70.1)		1079 (76.0)	3040 (79.5)	3088 (72.9)		999 (78.7)	2525 (80.1)	2945 (72.8)		3291 (75.5)	8607 (78.1)	8681 (72.0)	
**Marital status**				<0.001				0.274				0.865				<0.001
**In marriage**	630 (37.7)	1327 (32.8)	1195 (31.6)		207 (14.6)	535 (14.0)	554 (13.1)		46 (3.6)	106 (3.4)	134 (3.3)		883 (20.3)	1968 (17.9)	1883 (15.6)	
**Not in marriage**	1040 (62.3)	2722 (67.2)	2582 (68.4)		1212 (85.4)	3287 (86.0)	3680 (86.9)		1223 (96.4)	3047 (96.6)	3910 (96.7)		3475 (79.7)	9056 (82.1)	10172 (84.4)	
**Educational background**				0.035				0.004				0.513				<0.001
**Illiteracy**	960 (57.5)	2447 (60.4)	2310 (61.2)		950 (66.9)	2729 (71.4)	3015 (71.2)		1080 (85.1)	2693 (85.4)	3415 (84.4)		2990 (68.6)	7869 (71.4)	8740 (72.5)	
**Literacy**	710 (42.5)	1602 (39.6)	1467 (38.8)		469 (33.1)	1093 (28.6)	1219 (28.8)		189 (14.9)	460 (14.6)	629 (15.6)		1368 (31.4)	3155 (28.6)	3315 (27.5)	
**Living pattern**				<0.001				0.012				<0.001				<0.001
**With family members**	1397 (83.7)	3238 (80.0)	2959 (78.3)		1210 (85.3)	3142 (82.2)	3468 (81.9)		1151 (90.7)	2809 (89.1)	3516 (86.9)		3758 (86.2)	9189 (83.4)	9943 (82.5)	
**Alone or at nursing home**	273 (16.3)	811 (20.0)	818 (21.7)		209 (14.7)	680 (17.8)	766 (18.1)		118 (9.3)	344 (10.9)	528 (13.1)		600 (13.8)	1835 (16.6)	2112 (17.5)	
**Tobacco smoking status**				<0.001				0.002				<0.001				<0.001
**Non-smoker**	970 (58.1)	2422 (59.8)	2397 (63.5)		937 (66.0)	2687 (70.3)	3026 (71.5)		973 (76.7)	2537 (80.5)	3318 (82.0)		2880 (66.1)	7646 (69.4)	8741 (72.5)	
**Current smoker**	400 (24.0)	962 (23.8)	739 (19.6)		254 (17.9)	611 (16.0)	618 (14.6)		139 (11.0)	301 (9.5)	280 (6.9)		793 (18.2)	1874 (17.0)	1637 (13.6)	
**Former smoker**	300 (18.0)	665 (16.4)	641 (17.0)		228 (16.1)	524 (13.7)	590 (13.9)		157 (12.4)	315 (10.0)	446 (11.0)		685 (15.7)	1504 (13.6)	1677 (13.9)	
**Alcohol drinking status**				<0.001				<0.001				<0.001				<0.001
**Non-drinker**	1035 (62.0)	2579 (63.7)	2667 (70.6)		923 (65.0)	2635 (68.9)	3030 (71.6)		899 (70.8)	2310 (73.3)	3052 (75.5)		2857 (65.6)	7524 (68.3)	8749 (72.6)	
**Current drinker**	439 (26.3)	999 (24.7)	683 (18.1)		318 (22.4)	780 (20.4)	763 (18.0)		248 (19.5)	584 (18.5)	629 (15.6)		1005 (23.1)	2363 (21.4)	2075 (17.2)	
**Former drinker**	196 (11.7)	471 (11.6)	427 (11.3)		178 (12.5)	407 (10.6)	441 (10.4)		122 (9.6)	259 (8.2)	363 (9.0)		496 (11.4)	1137 (10.3)	1231 (10.2)	
**Regular exercise**				0.554				0.613				<0.001				0.002
**Yes**	763 (45.7)	1910 (47.2)	1780 (47.1)		553 (39.0)	1516 (39.7)	1634 (38.6)		368 (29.0)	1114 (35.3)	1313 (32.5)		1684 (38.6)	4540 (41.2)	4727 (39.2)	
**No**	907 (54.3)	2139 (52.8)	1997 (52.9)		866 (61.0)	2306 (60.3)	2600 (61.4)		901 (71.0)	2039 (64.7)	2731 (67.5)		2674 (61.4)	6484 (58.8)	7328 (60.8)	
**Meat consumption**				<0.001				<0.001				<0.001				<0.001
**Often**	702 (42.0)	1294 (32.0)	1131 (29.9)		624 (44.0)	1263 (33.0)	1316 (31.1)		599 (47.2)	1090 (34.6)	1311 (32.4)		1925 (44.2)	3647 (33.1)	3758 (31.2)	
**Occasionally**	685 (41.0)	2157 (53.3)	1828 (48.4)		597 (42.1)	2017 (52.8)	1982 (46.8)		496 (39.1)	1600 (50.7)	1748 (43.2)		1778 (40.8)	5774 (52.4)	5558 (46.1)	
**Rarely**	283 (16.9)	598 (14.8)	818 (21.7)		198 (14.0)	542 (14.2)	936 (22.1)		174 (13.7)	463 (14.7)	985 (24.4)		655 (15.0)	1603 (14.5)	2739 (22.7)	
**Fish consumption**				<0.001				<0.001				<0.001				<0.001
**Often**	421 (25.2)	594 (14.7)	586 (15.5)		437 (30.8)	631 (16.5)	753 (17.8)		416 (32.8)	507 (16.1)	767 (19.0)		1274 (29.2)	1732 (15.7)	2106 (17.5)	
**Occasionally**	781 (46.8)	2413 (59.6)	1864 (49.4)		622 (43.8)	2176 (56.9)	2023 (47.8)		547 (43.1)	1740 (55.2)	1699 (42.0)		1950 (44.7)	6329 (57.4)	5586 (46.3)	
**Rarely**	468 (28.0)	1042 (25.7)	1327 (35.1)		360 (25.4)	1015 (26.6)	1458 (34.4)		306 (24.1)	906 (28.7)	1578 (39.0)		1134 (26.0)	2963 (26.9)	4363 (36.2)	
**Vegetable consumption**				<0.001				<0.001				<0.001				<0.001
**Often**	1501 (89.9)	3474 (85.8)	3049 (80.7)		1232 (86.8)	3158 (82.6)	3221 (76.1)		1086 (85.6)	2528 (80.2)	2945 (72.8)		3819 (87.6)	9160 (83.1)	9215 (76.4)	
**Occasionally**	133 (8.0)	502 (12.4)	581 (15.4)		149 (10.5)	541 (14.2)	723 (17.1)		143 (11.3)	502 (15.9)	729 (18.0)		425 (9.8)	1545 (14.0)	2033 (16.9)	
**Rarely**	36 (2.2)	73 (1.8)	147 (3.9)		38 (2.7)	123 (3.2)	290 (6.8)		40 (3.2)	123 (3.9)	370 (9.1)		114 (2.6)	319 (2.9)	807 (6.7)	
**Fruits consumption**				<0.001				<0.001				<0.001				<0.001
**Often**	556 (33.3)	924 (22.8)	789 (20.9)		513 (36.2)	881 (23.1)	961 (22.7)		494 (38.9)	799 (25.3)	1043 (25.8)		1563 (35.9)	2604 (23.6)	2793 (23.2)	
**Occasionally**	686 (41.1)	2109 (52.1)	1822 (48.2)		543 (38.3)	1920 (50.2)	1880 (44.4)		464 (36.6)	1492 (47.3)	1680 (41.5)		1693 (38.8)	5521 (50.1)	5382 (44.6)	
**Rarely**	428 (25.6)	1016 (25.1)	1166 (30.9)		363 (25.6)	1021 (26.7)	1393 (32.9)		311 (24.5)	862 (27.3)	1321 (32.7)		1102 (25.3)	2899 (26.3)	3880 (32.2)	
**Tea drinking**				<0.001				<0.001				<0.001				<0.001
**Often**	553 (33.1)	1233 (30.5)	1022 (27.1)		492 (34.7)	993 (26.0)	935 (22.1)		349 (27.5)	592 (18.8)	699 (17.3)		1394 (32.0)	2818 (25.6)	2656 (22.0)	
**Occasionally**	258 (15.4)	680 (16.8)	536 (14.2)		193 (13.6)	700 (18.3)	521 (12.3)		153 (12.1)	624 (19.8)	445 (11.0)		604 (13.9)	2004 (18.2)	1502 (12.5)	
**Rarely**	859 (51.4)	2136 (52.8)	2219 (58.8)		734 (51.7)	2129 (55.7)	2778 (65.6)		767 (60.4)	1937 (61.4)	2900 (71.7)		2360 (54.2)	6202 (56.3)	7897 (65.5)	
**Hypertension**				0.008				0.031				0.605				0.005
**Yes**	956 (57.2)	2302 (56.9)	2272 (60.2)		759 (53.5)	2078 (54.4)	2404 (56.8)		649 (51.1)	1647 (52.2)	2133 (52.7)		2364 (54.2)	6027 (54.7)	6809 (56.5)	
**No**	714 (42.8)	1747 (43.1)	1505 (39.8)		660 (46.5)	1744 (45.6)	1830 (43.2)		620 (48.9)	1506 (47.8)	1911 (47.3)		1994 (45.8)	4997 (45.3)	5246 (43.5)	
**Heart disease**				0.015				0.007				0.040				<0.001
**Yes**	147 (8.8)	282 (7.0)	320 (8.5)		95 (6.7)	209 (5.5)	304 (7.2)		73 (5.8)	147 (4.7)	243 (6.0)		315 (7.2)	638 (5.8)	867 (7.2)	
**No**	1523 (91.2)	3767 (93.0)	3457 (91.5)		1324 (93.3)	3613 (94.5)	3930 (92.8)		1196 (94.2)	3006 (95.3)	3801 (94.0)		4043 (92.8)	10386 (94.2)	11188 (92.8)	
**Cerebrovascular disease**				0.103				0.613				0.775				0.287
**Yes**	94 (5.6)	175 (4.3)	175 (4.6)		57 (4.0)	137 (3.6)	168 (4.0)		35 (2.8)	99 (3.1)	119 (2.9)		186 (4.3)	411 (3.7)	462 (3.8)	
**No**	1576 (94.4)	3874 (95.7)	3602 (95.4)		1362 (96.0)	3685 (96.4)	4066 (96.0)		1234 (97.2)	3054 (96.9)	3925 (97.1)		4172 (95.7)	10613 (96.3)	11593 (96.2)	
**Respiratory disease**				0.947				0.898				0.860				0.928
**Yes**	217 (13.0)	537 (13.3)	493 (13.1)		153 (10.8)	397 (10.4)	450 (10.6)		114 (9.0)	297 (9.4)	384 (9.5)		484 (11.1)	1231 (11.2)	1327 (11.0)	
**No**	1453 (87.0)	3512 (86.7)	3284 (86.9)		1266 (89.2)	3425 (89.6)	3784 (89.4)		1155 (91.0)	2856 (90.6)	3660 (90.5)		3874 (88.9)	9793 (88.8)	10728 (89.0)	
**Cognitive impairment**				0.023				<0.001				<0.001				<0.001
**Yes**	217 (13.0)	529 (13.1)	568 (15.0)		471 (33.2)	1272 (33.3)	1599 (37.8)		743 (58.6)	1836 (58.2)	2548 (63.0)		1431 (32.8)	3637 (33.0)	4715 (39.1)	
**No**	1453 (87.0)	3520 (86.9)	3209 (85.0)		948 (66.8)	2550 (66.7)	2635 (62.2)		526 (41.4)	1317 (41.8)	1496 (37.0)		2927 (67.2)	7387 (67.0)	7340 (60.9)	
**ADL disability**				0.140				<0.001				<0.001				<0.001
**Yes**	526 (31.5)	1243 (30.7)	1238 (32.8)		662 (46.7)	1648 (43.1)	2105 (49.7)		785 (61.9)	1883 (59.7)	2706 (66.9)		1973 (45.3)	4774 (43.3)	6049 (50.2)	
**No**	1144 (68.5)	2806 (69.3)	2539 (67.2)		757 (53.3)	2174 (56.9)	2129 (50.3)		484 (38.1)	1270 (40.3)	1338 (33.1)		2385 (54.7)	6250 (56.7)	6006 (49.8)	

BMI: body mass index; ADL: activity of daily living.

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
