# Peer review of "Garlic Consumption and All-Cause Mortality among Chinese Oldest-Old Individuals: A Population-Based Cohort Study"

_nutrients, 2019, doi:10.3390/nu11071504_

Round 1
Reviewer 1 Report
This paper investigated the associations between garlic consumption and all-cause mortality among Chinese Oldest-Old subjects. Overall the manuscript is well written and the findings are really interesting. Please check all the abbreviations and acronyms, and define them in the text. The statistical analysis has been well carried out. Tables and Figures are clear and exhaustive.
I would suggest to include more references in the discussion, in order to emphasize the anticancer properties of garlic and compare them to other substances with similar effects (e.g. Vacante M et al, Nutrients 2019). I would also include a statement in the conclusion paragraph on the possible future clinical applications of the results of the study.
Author Response
Response to Reviewer 1 Comments
This paper investigated the associations between garlic consumption and all-cause mortality among Chinese Oldest-Old subjects. Overall the manuscript is well written and the findings are really interesting. The statistical analysis has been well carried out. Tables and Figures are clear and exhaustive.
Point 1: Please check all the abbreviations and acronyms, and define them in the text.
Response 1: Thank you for pointing these out. To ensure all the abbreviations and acronyms were properly defined, we checked it thoroughly again and addressed the existed problems. Abbreviations defined in the revised manuscript are also listed in Table A1.
Table A1. List of abbreviations
ADL | activities of daily living |
BMI | body mass index |
CLHLS | the Chinese Longitudinal Healthy Longevity Study |
CVD | cardiovascular disease |
HRs | hazard ratios |
MMSE | the mini-mental state examination |
OSCs | organosulfur compounds |
OxLDL | oxidized low density lipoprotein |
ROSs | reactive oxygen species |
Point 2: I would suggest to include more references in the discussion, in order to emphasize the anticancer properties of garlic and compare them to other substances with similar effects (e.g. Vacante M et al, Nutrients 2019)
Response 2: Anticancer properties of garlic has been further addressed and compared them to other substances with similar effects (eg. olive oil) in the discussion in our revised manuscript. “Garlic and cancer: Based on a study by the US Food and Drug Administration and a Chinese double-blind intervention study, a number of investigations on humans have revealed the anti-carcinogenic potential of garlic (Ref:Kim, J.Y.; Kwon, O. Garlic intake and cancer risk: an analysis using the Food and Drug Administration’s evidence-based review system for the scientific evaluation of health claims. Am J Clin Nutr 2009, 89, 257-264. Li, H.; Li, H.Q.; Wang, Y.; Xu, H.X.; Fan, W.T.; Wang, M.L.; Sun, PH.; Xie, X.Y. An intervention study to prevent gastric cancer by microselenium and large dose of allitridum. Chin Med J (Engl) 2004, 117, 1155-1160.). The garlic extract has been reported to inhibit growth of several different cancer cells in vitro, as well as cancer growth in vivo, enhancing the activities of chemotherapeutics, as well as MAPK and PI3K inhibitors (Ref:Petrovic, V.; Nepal, A.; Olaisen, C.; Bachke, S.; Hira, J.; Søgaard, C.; Røst, L.; Misund, K.; Andreassen, T.; Melø, T.; et al. Anti-Cancer Potential of Homemade Fresh Garlic Extract Is Related to Increased Endoplasmic Reticulum Stress. Nutrients 2018, 10, 450.). Besides, similar to extract of other substances like olive oil, garlic may also have an active effect on chemopreventive action due to its phenolic compounds (Ref: Borzì, A.; Biondi, A.; Basile, F.; Luca, S.; Vicari, E.; Vacante, M. Olive Oil Effects on Colorectal Cancer. Nutrients 2019 11, 32.).”
Point 3: I would also include a statement in the conclusion paragraph on the possible future clinical applications of the results of the study.
Response 3: Thank you for your good suggestion. Previous studies have pointed out that homemade ethanol-based garlic extract can inhibit growth of several different cancer cells in vitro, as well as cancer growth in vivo in a syngeneic orthotopic breast cancer model. In our study, we provided evidence on the association of garlic consumption with decreased all-cause mortality among the oldest old. Further experiments should be conducted to verify which constituent was most essential to reduce the risk of mortality. We added in the “Conclusions” part that:” The garlic constituent can be further studied to provide effective agents for its potential use in anti-inflammation, anti-cancer treatment and CVD prevention.”
Reviewer 2 Report
This is an interesting prospective cohort study examining the association between garlic consumption and all-cause mortality in Chinese adults aged 80 and above. However, there are several major issues which need addressing:
-The study is described as a prospective cohort but garlic consumption at age 60 was collected retrospectively. This variable is therefore subject to recall bias, especially at such older ages.
-The quantity of garlic consumed was not assessed (only frequency) and this is a major source of bias.
- It is likely there is a substantial amount of residual confounding. Models were adjusted for frequency of eating vegetables, meat, fish, and drinking tea. However, this dietary adjustment seems very limited. What about adjusting for overall dietary patterns, fruit intake, total energy intake, intake of vitamins and minerals etc.
- There are also other sources of potential confounding such as income, occupational socioeconomic status, BMI, hypercholesterolemia, medication use (e.g. statins, beta-blockers etc.). The authors should adjust for these additional confounding factors or mention them more thoroughly in the limitations section of the discussion.
Minor comments:
Methods
-Study design and participants – please can the authors provide p values for the comparisons between those lost-to-follow up or not.
-Has the FFQ used been validated previously?
-How representative is the CLHLS of the general Chinese population, and therefore how generalizable are the results?
Results
-It would be interesting to perform a sub-analysis of mortality by type. As a bare minimum, splitting into CVD and cancer mortality would be useful.
References
-Please can the authors check ref 9 – I think the year should be 2008 not 2013.
Author Response
Dear reviewer of Nutrients,
Thank you for providing detailed informative comments and suggestions, which helped to improve our manuscript entitled “Garlic Consumption and All-Cause Mortality Among Chinese Oldest-Old Individuals: A Population-Based Prospective Cohort Study” (Manuscript ID: nutrients-519137). We have made revisions according to your comments and provided a detailed point-by-point response in the response letters.
Attached please find the “author-coverletter-4360057.v3.docx”. We greatly appreciate your reconsideration of our manuscript.

Round 2
Reviewer 2 Report
Thank you for addressing all of my concerns and comments adequately.